# Simultaneity Factors of Public Electric Vehicle Charging Stations Based on Real-World Occupation Data

Christopher Hecht [1,2,3,*], Jan Figgener [1,2,3] and Dirk Uwe Sauer [1,2,3,4]

1   Grid Integration and Storage System Analysis, Institute for Power Electronics and Electrical Drives (ISEA), RWTH Aachen University, 52074 Aachen, Germany; jan.figgener@isea.rwth-aachen.de (J.F.); dirkuwe.sauer@isea.rwth-aachen.de (D.U.S.)
2   Institute for Power Generation and Storage Systems (PGS), E.ON ERC, RWTH Aachen University, 52074 Aachen, Germany
3   Juelich Aachen Research Alliance, JARA-Energy, 52056 Aachen, Germany
4   Helmholtz Institute Muenster (HI MS), IEK-12, Forschungszentrum Jülich, 52428 Jülich, Germany
*   Correspondence: christopher.hecht@isea.rwth-aachen.de or batteries@isea.rwth-aachen.de; Tel.: +49-241-80-49366

**Abstract:** Charging of electric vehicles may cause stress on the electricity grid. Grid planners need clarity regarding likely grid loading when creating extensions. In this paper, we analyse the simultaneity factor (SF) or peak power of public electric vehicle charging stations with different recharging strategies. This contribution is the first of its kind in terms of data quantity and, therefore, representativeness. We found that the choice of charging strategy had a massive impact on the electricity grid. The current "naive" charging strategy of plugging in at full power and recharging until the battery is full cause limited stress. Price-optimised recharging strategies, in turn, create high power peaks. The SFs varied by strategy, particularly when using several connectors at once. Compared to the SF of a single connector in naive charging, the SF decreased by approximately 50% for groups of 10 connectors. For a set of 1000 connectors, the SF was between 10% and 20%. Price-optimised strategies showed a much slower decay where, in some cases, groups of 10 connectors still had an SF of 100%. For sets of 1000 connectors, the SF of price-optimised strategies was twice that of the naive strategy. Overall, we found that price optimisation did not reduce electricity purchase costs by much, especially compared to peak-related network expansion costs.

**Keywords:** electric vehicle; charging station; simultaneity factor; charging strategy; electricity grid; peak power





## 1. Introduction

Electric vehicles (EVs) are gaining significant market shares around the world [1]. This has led to additional electricity consumption and may cause, if not conducted intelligently, spikes in power consumption. At the same time, this addition of new vehicles is, however, also an opportunity for flexibility, since EVs spend significantly longer durations connected to charging stations than they do actually charging [2]. By shifting the electricity consumption of EVs, several goals may be pursued including [3]:

- Cost efficiency:
  Vehicles optimize their charging behaviour to charge during the hours of the day when electricity is available at the cheapest price;
- Limiting power demand from the grid:
  Vehicles reduce their demand for charging power during times of high grid usage in order to avoid overburdening the electricity grid;
- Usage of intermittent renewable energy:

Vehicles try to satisfy their energy demand, as much as possible, through the use of intermittent renewable energies such as PV and wind.

- Increasing the system's flexibility:

Increase the flexibility of the system, for example, via frequency control, power quality management, or use of backup power.

The goal of this paper was to estimate the potential benefits for Germany of the first three goals.

### 1.1. Literature Review

Since the number of EVs is experiencing exponential growth in many countries around the world [1], the technology has attracted significant interest within the public sphere. As Figure 1 shows, the same is true for academic interest on the topic, resulting in an increasing number of publications.

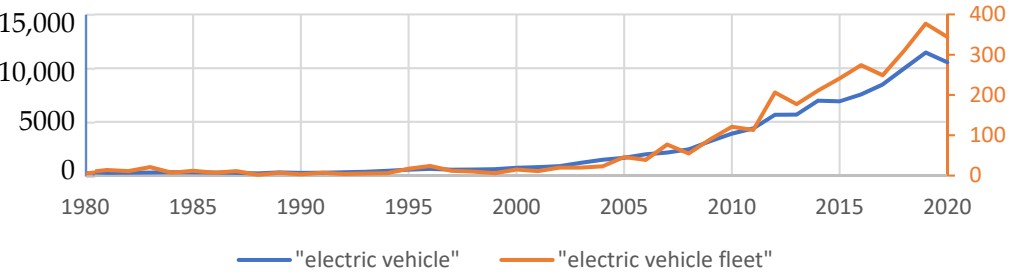

**Figure 1.** Number of publications found via Scopus® per year for the search terms shown in the legend. Note that not all of the publications submitted in 2020 will likely have been accepted yet, leading to a slightly lower actual value for 2020 [4].

One key topic when discussing EVs is the understanding of how and when their recharging occurs and how it can best be managed to reduce the negative impacts on the grid [5–12]. This is often conducted from the perspective of trying to optimise the charging behaviour of an aggregation of EVs. The methods employed include numerical optimisation based on known or predicted arrival and departure times [5,6] and data-driven methods such as reinforcement learning [7,8]. Both variants require an optimisation target, such as electricity prices [7–10], often coupled with constraints to ensure that battery capacity limits and charging power limits are respected and that grid assets are not overloaded. Some studies also focused on how additional devices, such as photovoltaics installations, could be used for the same purpose [13,14].

#### 1.1.1. Simultaneity Factor (SF)

While there are many different contributions of the aforementioned types, bird's-eye view perspectives focussing on distribution and transmission grids are harder to find. Some examples that calculate an SF are in References [12,15–20] (although [17,18] use mostly the same data, and [20] is only available with limited access). Field tests have also been executed in this regard [21], but the small number of vehicles makes the results challenging to generalise. Here, the SF can be understood as the maximum occurring power divided by the sum of the installed charging station power as a function of the number of charging stations under observation. Due to the current lack of real-world data and the novelty of the technology, most of the results are based on simulations of charging station usage [22–24] (see also the literature review in these papers) or statistical averaging. In contrast to these simulations, this paper was based on real-world data.

Figure 2 shows the SFs found in [12,18]. The SF is defined, in this context, in terms of the sum of electric vehicle charging power as outlined in the equation below. As can be seen, the estimates varied strongly between the two sources, likely due to the many assumptions that need to be made.

$$SF_{vehicles} = \frac{P_{max}}{P_{sum,vehicles}} \tag{1}$$

where $SF_{vehicles}$ is the SF defined in terms of vehicles; $P_{max}$ is the instantaneous maximum power recorded; $P_{sum,vehicles}$ is the sum of the charging power of all observed vehicles.

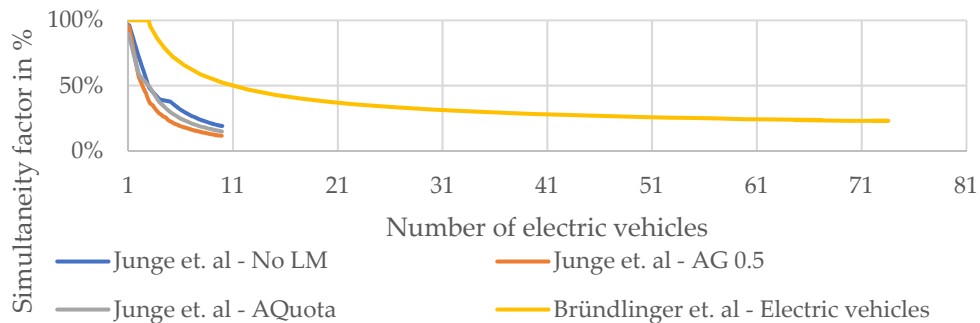

**Figure 2.** SFs reported in the literature by Junge et al. [12] (three scenarios) and by Bründlinger et al. [18] for electric vehicle charging. An SF of 100% was equivalent to all observed vehicles charging simultaneously at one moment in time. For a description of the individual scenarios, please refer to the linked studies.

Figure 3 shows the SFs as reported in [16]. In contrast to the previous vehicle-based SFs, in this case, the reference power was the sum of the installed connector capacity ($P_{sum,connectors}$) as shown in formula below. As can be seen, a wide gap exists between connectors with and without flexible energy tariffs. Understanding this difference is a key topic of this paper. Unfortunately, to the authors' best knowledge, there is no other publication performing this type of calculation, even though it is crucial for grid planning.

$$SF_{connectors} = \frac{P_{max}}{P_{sum,connectors}} \tag{2}$$

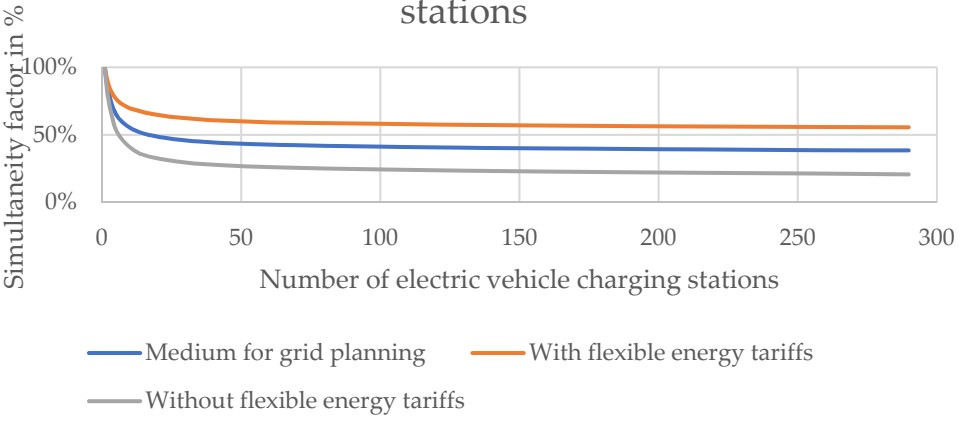

**Figure 3.** SFs reported in the literature [16] for electric vehicle charging stations. An SF of 100% is equivalent to all observed charging stations having a vehicle charging at full power at one moment in time.

1.1.2. Impact on Electricity Prices and Emission Intensity

One of the most discussed topics in regard to the recharging strategies of EVs is what economic benefit can be generated from smart charging [25–28]. These calculations are

typically performed on a per vehicle and year level and result in 200–500 EUR/a for the optimistic case of perfect foresight and unidirectional charging on several markets.

For the system's impact, which is more central to this paper, Reference [3] offers a comprehensive review of which a short excerpt is shown in Table 1. Particularly scenario "V1G" is comparable to the calculations made in this paper. A key difference, however, is that the authors applied a mixed strategy, this paper we analysed the impact of individual strategies. In the long term, the authors expect that smart charging will be able to significantly reduce marginal costs and $CO_2$ emissions, particularly for PV-based systems.

**Table 1.** Short-term impact of EV charging on the selected key performance indicators. BAU = Business as usual; V1G = smart unidirectional charging; V2G = bidirectional charging; MaaS = mobility as a service. Taken from [3].

|  | Curtailment | Δ Peak Load (%) | Δ Marginal Cost (%) | Δ in $CO_2$ Emissions (%) |
|---|---|---|---|---|
| BAU | 2% | | | |
| V1G | 1% | −3% | −1% | −1% |
| V2G | | −4% | −13% | −2% |
| MaaS | | 8% | −8% | 14% |

### 1.1.3. Our Contribution

With this work, we aim to support grid planners and energy utility companies as well as researchers in academia in the same fields in understanding impact of EVs on grids. Specifically, we used large-scale real-world charging station occupation data to calculate a representative sample of SFs under a variety of conditions. The only other publication to have done so extensively is [16] to the best of our knowledge. The aforementioned document was published in February 2018, and since the EV market is highly dynamic, an update as well as a verification are crucial for the integration of electric vehicles into the public electricity grid. We provide this information using representative data for Germany.

We performed this calculation using various strategies typically suggested in the literature to offer a view of their impact on grid utilisation. The obtained information is critical for grid infrastructure expansions, since grid assets, such as cables and transformers, have very long lifetimes of 40 years or more. It must therefore be prevented that a cable is replaced earlier than necessary to make full use of the investment. Our contributions in short are:

1. Calculation of SF using real-world charging station occupation data;
2. Comparison of the grid impacts of various EV recharging strategies;
3. Energy cost comparison when employing these strategies;
4. Comparison of emission intensities when employing these strategies.

The rest of this paper is structured as follows. First, the data used for the ensuing analysis are presented in Section 2. Next, the methodology is introduced in Section 3, where first a focus is laid on the algorithms used and then how this can be combined with the data into strategies. The results of running the strategies are shown in Section 4 with a focus on the four core contributions given above. In Sections 5 and 6, we discuss these results and draw conclusions for practitioners and researchers in the fields of electromobility and grids.

## 2. Data

This paper is based on the data presented in our previous publication [2]. For the readers' convenience, the key aspects are repeated here.

The data were collected by regularly accessing the maps of several online charging station roaming maps. Since these maps cover most of Germany, 26,951 charging station connectors could be observed between 21 December 2019 and 10 March 2020. The timeframe was chosen to avoid the impact of the first lockdown due to the coronavirus in 2020. While it is difficult to estimate how many connectors there are in Germany, it can be assumed that

the used data are representative of the country, since, at the time, the German federal grid agency reported 25,434 connectors [29] and ChargeMap reported 61,665 connectors [30].

To keep the calculation time within a reasonable timeframe, a random subsample was taken from the original data of which the locations are shown in Figure 4. When creating the subsample, we ensured that all connectors were used at least once in the observed period and allowed for connectors near the border to be part of the dataset as well. The resulting dataset contained 1562 connectors with 45,487 charge events (CEs) recorded.

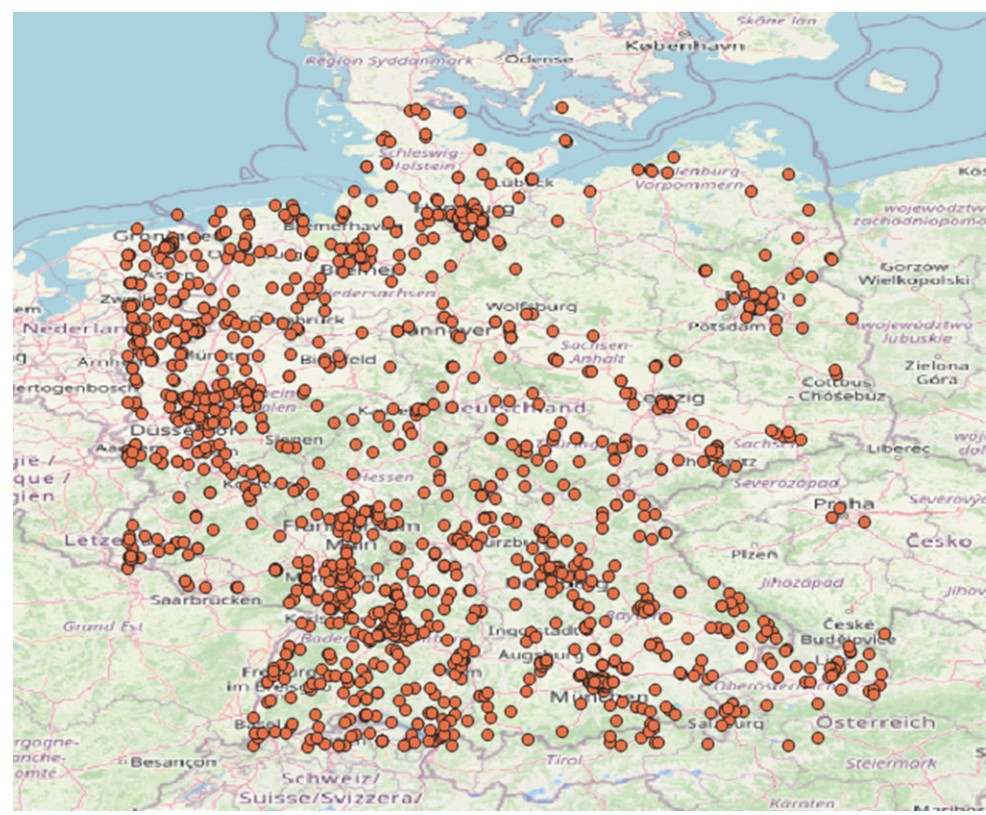

**Figure 4.** Locations of the connectors used in this study. Created with [31].

Concerning usage patterns, it was observed that, especially, AC chargers were occupied much more during the day than during the night. This could be explained partially by people arriving at work between 6 a.m. and 9 a.m. and leaving their car at the charging station afterwards. The time spent at a car at a connector was virtually identical for stations with 11 kW and 22 kW power. This is a clear hint that the time spent at the station is influence mostly by other factors than the actual charging needs. Given the topic of this paper, this offers the potential to estimate the flexibility in charging processes. A copy of the overview of the connector occupation is further given in Figure 5.

The other data sources used in this paper were the day ahead prices for Germany [32] and the $CO_2$ intensity of the German electricity grid [33]. These two datasets are visualised in Figure 6. Figure 6a,b show how the two variables behaved over the time observed in this study and on a weekly average, respectively. Figure 6c displays the correlation between the two variables. It can be seen that the two were correlated, which was to be expected, since photovoltaics and wind power are both free of emissions and have marginal costs of 0.

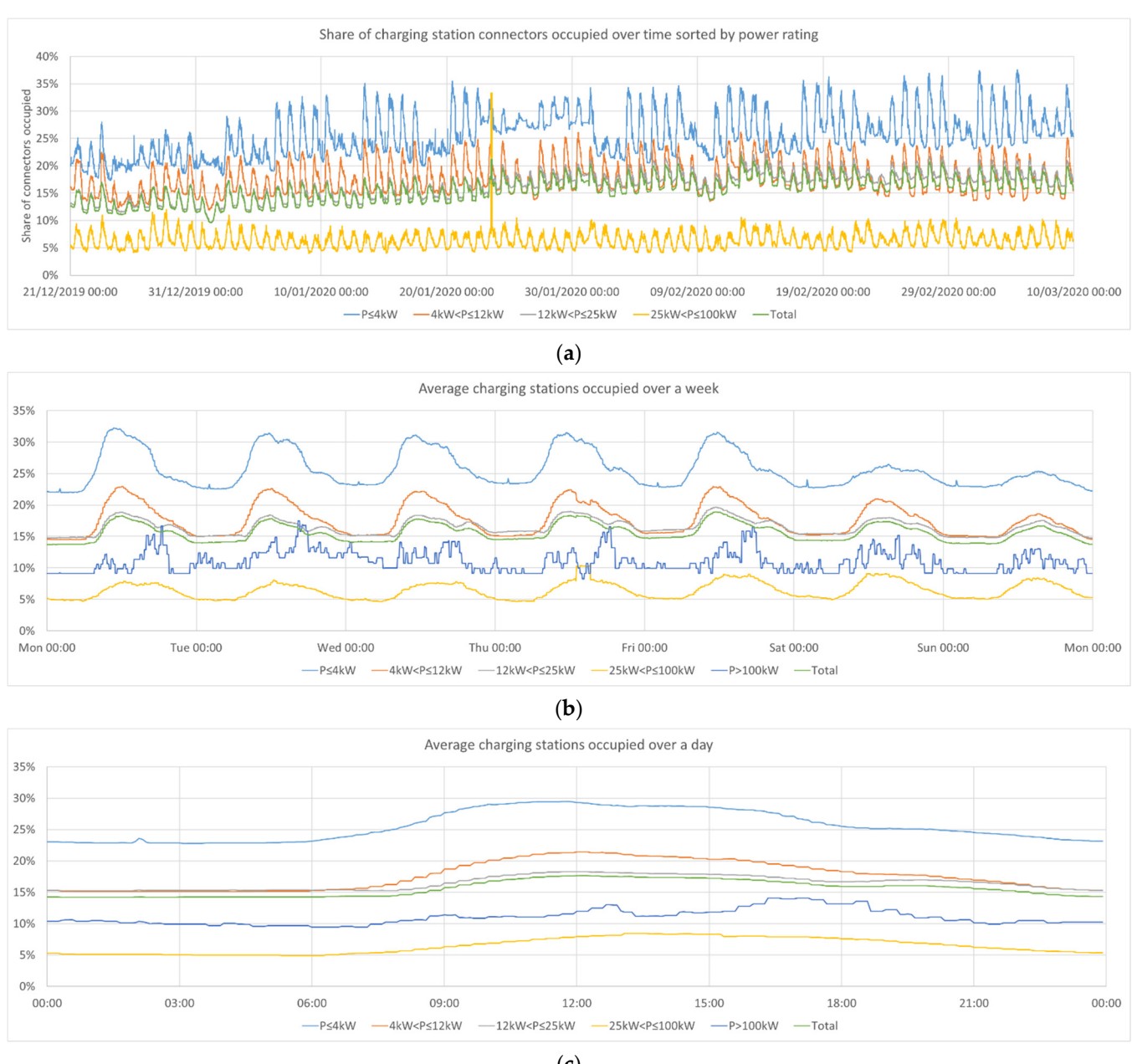

(**a**)

(**b**)

(**c**)

**Figure 5.** Share of the charging station connectors that were occupied at each moment in time by power rating. For each power rating, the number of connectors marked as occupied was divided by the sum of the connectors for which the status was known: (**a**) the occupation rate over the entire observed period; (**b**) aggregated on a weekly level; (**c**) aggregated on a daily level. Note that the peak on 23 January 2020 as well as the small peaks at approximately 2:10 in the morning were reported by the observed websites but likely represent a fault in the IT system on the remote end. The data and their description are a direct copy from our previous publication and repeated here for the reader's convenience [2] (This article was published in *eTransportation*, volume 6, Hecht, C.; Das, S.; Bussar, C.; Sauer, D.U., Representative, empirical, real-world charging station usage characteristics and data in Germany, 100079, Copyright Elsevier 2020).

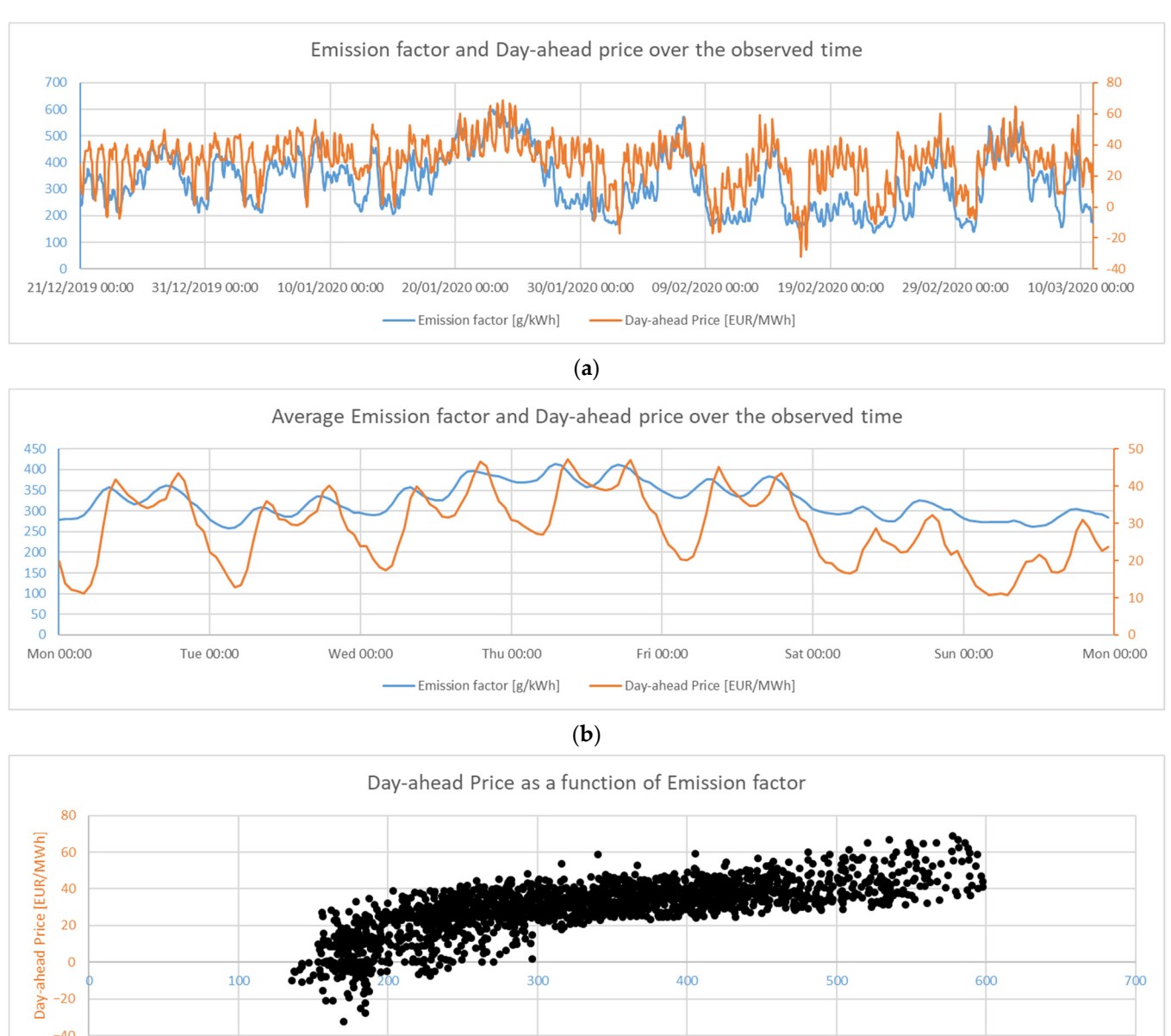

**Figure 6.** Emission factor [33] and day-ahead prices [32] used in this paper: (**a**) the full analysed period; (**b**) the average over a week; (**c**) the relationship between the two factors. In the scatter plot in (**c**), each dot reflects one hour, where the *x*- and *y*-axis values are defined by the mentioned prices and emission factors.

## 3. Methodology

The goal of this paper was to show how different charging strategies affect peak power and the simultaneity factor when using public electric vehicle charging stations. To achieve this, these strategies must be implemented in a computationally efficient manner, as a relatively large dataset was analysed in this paper. These algorithms need to be able to use the prices and $CO_2$ intensity of the electricity mix to form strategies for when actual recharging should happen when a vehicle is connected to a station.

In this chapter, the methodology is explained in a two-step approach. First, the algorithms developed by the authors are outlined without yet considering the input data and conditions. In the second step, these algorithms are combined with the input data,

such as energy requirements per charging process, and power limits, to form strategies that form the basis for the discussions in the rest of this paper.

### 3.1. Optimization Algorithms

Three distinct algorithms were developed and used on the 1562 connectors between 21 December 2019 and 10 March 2020 as outlined in Section 2. All calculations were performed on an hourly resolution but could, in principle, be performed on shorter timescales as well. The reason for choosing an hourly resolution was that it offered a reasonable balance between calculation time and accuracy and because the price and emission intensity values are given on an hourly resolution as well. These algorithms can be found in the following subchapters.

### 3.1.1. Naive Algorithm

The naive algorithm mirrors the behaviour of vehicles in today's public charging infrastructure, where arriving vehicles charge at stations with available power until the battery is completely full. Since no information regarding the vehicle is available, the algorithm considers only the power limit posed by the connector.

### 3.1.2. Price-Optimisation Algorithm

The goal of the price-optimisation algorithm is to schedule the charging of vehicles at the connectors such that the energy purchase price, as given by the day-ahead prices, is minimised. This aim can mathematically be formulated as follows:

$$\min_{\lambda} \sum_{i=0}^{n} P_{i,k_i} \cdot \lambda_{k_i} \, \text{s.t.} \sum_{k=j}^{T_{tot,\,i}} P_{i,k_i} \cdot T = E \cdot n - e \; \forall i \tag{3}$$

where n is the number of recorded charge events; $P_{i,ki}$ is the charging power in charge event i at time $k_i$; $\lambda_{ki}$ is the cost of recharging at time $k_i$; T is the optimization time step, chosen to be one hour; $T_{tot,i}$ is the duration of the charge event i; E is the energy consumption assumed per charge event; e is an error term that reduces E·n to compensate so that not all events are long enough to supply planned E.

The objective was achieved by starting with the cheapest time step of the observed period and scheduling connectors to which a car that was not yet fully charged was connected in order to charge during that time step. This process was iteratively repeated for all time steps ordered by price. The process is outlined in the following list and the pseudo-code of the algorithm is given in Figure A1 (see Appendix A). A sample output is given in Figure 7.

1.  Sort the price time series by price in ascending order;
2.  Iterate through the sorted time series, and perform the following steps for each timestamp, ts:
    a.  Obtain the charge events (CE) happening at ts;
    b.  Obtain the energy already charged per CE;
    c.  Calculate energy that still needs to be charged per CE;
    d.  Iterate through the list found in (a), and charge each event by the maximum of the station power limit or the remaining energy found in (c).

Due to the iterative approach, the algorithm always detects the global minimum for charging costs. The approach can also be parallelized, since the individual events are independent of each other.

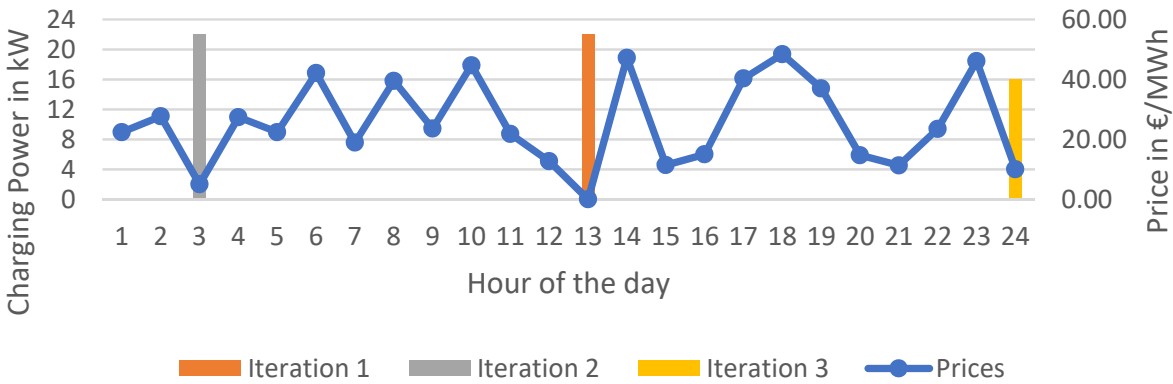

**Figure 7.** Example of the price-optimisation algorithm. The prices are illustrative only. Iterations 1 to 3 show the charging power that the algorithm would assign to each hour assuming that the vehicle was connected for 24 h, required 60 kWh, and was able to recharge at 22 kW. Charging was first assigned to hour 13, as the price was the lowest. The next-lowest price was at hour 3, and the algorithm decided to also charge with full power during that hour. Since 44 kWh were already charged, only 16 remained to be charged at the next-cheapest hour, 24.

### 3.1.3. CO$_2$-Optimization Algorithm

The CO$_2$-optimization algorithm is almost identical to the price-optimisation algorithm, with the key difference being that instead of price, the hourly CO$_2$ emissions are used as the target. The time series was consequently sorted by grid CO$_2$ intensity instead of energy prices, and the least emission-intense hours were used where possible to perform recharging. Due to the similar approach, a detailed explanation is omitted.

### 3.1.4. Peak Power Reduction Algorithm

The goal of the peak power reduction algorithm is to take a planned schedule, such as the output of the naive algorithm or the price-optimisation algorithm, and ensure that the peak power remains below a given threshold. The goal can be expressed though the following equation:

$$\min_\lambda \sum_{i=0}^{n} P_{i,k_i} \cdot \lambda_{k_i} \, \text{s.t.} \sum_{k=0}^{T_{tot,\,i}} P_{i,k_i} \cdot T = E \cdot n - e \; \forall i \wedge \max\left(\sum_{k=0}^{1921} P_k\right) = P_{max} \tag{4}$$

where the equation is almost identical to the one used in Section 3.1.2, with the additional condition that the sum of the charging station power across all connectors for all 1921 h observed in this dataset may not exceed a defined peak power, $P_{max}$. The criterion is, however, a weak condition and may be violated in specific hours if no solution is otherwise found.

The chosen implementation was to check for the hour with the highest power and whether any charger could also charge at another point in time. The cheapest alternatives were selected, and the charging moved. This process was repeated until it was either no longer possible to improve the time steps or until the highest power demand was at or below the threshold value. The process is outlined in the following list and the pseudo-code of this algorithm is given in Figure A2. A sample output is given in Figure 8.

1. Create an empty list list$_{ts}$ of timestamps, ts;
2. While the length of list$_{ts}$ does not contain all ts do the following:

   (a) Find the ts with the highest power flow;
   (b) If the power flow is less than the power limit, terminate the algorithm;
   (c) Obtain CEs occurring at ts;
   (d) For each CE, check if the charging process can also be performed at another moment in time. If this is possible, shift the charging to the least critical moment in time, starting with the cheapest option;

(e)     If there is no option to perform the charging at another moment in time or ts has been attempted n times already, add ts to list_{ts}.

In principle, the algorithm above would be able to find a global minimum. The challenge is that as the number of ts that experience peak power (i.e., the peak becoming flatter), the incremental improvements per iteration eventually become very small. For this reason, n was chosen as 100 to balance runtime and accuracy based on sensitivity analyses.

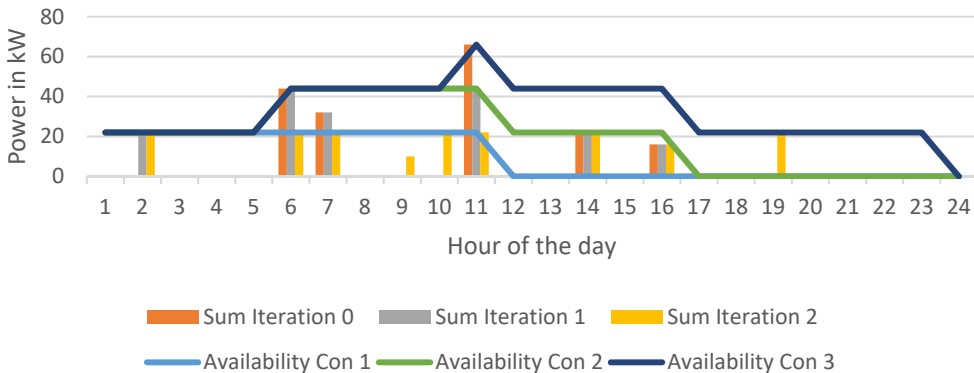

**Figure 8.** Example of the peak power reduction algorithm. Legend entries starting with "Sum" show the sum of the power demand of the three observed connecters for each iteration. Labels starting with "Availability" show the difference between the sum of the charging power of all connected vehicles and the already scheduled recharges at that instance.

### 3.1.5. Comparison to Algorithms in Literature

We designed the outlined algorithms for speed and reproducibility. Speed was necessary, since runtimes would otherwise be excessive given the large dataset. In this subchapter, we briefly compare our approach to other possibilities as reported in Section 1.1.

A numerical optimiser would have to select the optimal charging moments for over 45,000 CEs, each lasting up to days. If the power to charge during each hour that a vehicle was connected was used as the optimisation variable, this would introduce 100,000 s of such variables. Such an optimisation would have excessive runtimes and not necessarily converge to a global maximum. The price-optimisation algorithm in this paper was able to solve the problem in a reasonable runtime while being guaranteed to reach a global maximum.

A neural network or alike methods were not necessary in this case, since perfect knowledge over the entire observed period is assumed. Removing this assumption would require some kind of forecasting methodology which, in turn, creates a level of randomness and uncertainty with regards to the results presented. Since the objective of this paper was not to introduce smart recharging techniques but rather to evaluate overall system impact, this approach did not seem feasible.

### 3.2. Strategies

In this paper, four strategies are referred to as follows:

- Naive strategy: corresponds to the naive algorithm outlined in Section 3.1.1;
- Price-optimisation strategy: corresponds to price-optimisation algorithm in Section 3.1.2;
- Emission reduction strategy: usage of the price-optimisation algorithm with $CO_2$ intensity instead of day-ahead prices as optimisation target;
- Peak minimisation strategy: combination of price-optimisation algorithm (Section 3.1.2 and peak power reduction algorithm Section 3.1.4).

For the peak power reduction strategy, the maximum peak power was calculated as follows. The values were chosen based on own preliminary studies, where the upper limit showed that barely any peak power reduction was necessary (i.e., the peak without peak limitation was already very close to the upper limit). The lower limit of zero represented

an extreme and unachievable goal for the algorithm. The optimization will run until it has attempted each crtical_ts in Figure A2 for 100 times. The latter value was chosen as a compromise between accuracy and runtime.

$$P_{max} = \{x : 704 \text{ kW} * a \wedge a \in [0, 1, \ldots 10]\} \tag{5}$$

Each strategy was run for an assumed energy demand per vehicle of 8, 20, 40, and 60 kWh. These values were chosen based on the arguments given in [2] and repeated here for convenience. The 8 kWh corresponded to [34], but since the document reported on the period 2012–2016, higher values for energy demand were also considered. Most Evs require 15 kWh/100 km and 25 kWh/100 km [35], and cars in Germany are driven, on average, 39 km/day [36]. In addition, 20 kWh was a lower bound that represented two recharges per week for an efficient vehicle, and 60 kWh resulted in a single recharge per week, even for an inefficient vehicle. With the current vehicle fleet, this value was unrealistically high and should therefore be understood as an upper bound, at least for the next few years.

## 4. Results

In the following, the results of applying the strategies are outlined along four key dimensions, namely, peak power, electricity costs, $CO_2$ intensity, and SF.

### 4.1. Power Demand Shape

Figure 9 shows the power demand resulting from using the naive strategy. It can be seen that the power demands were much more regular over the course of a week as compared to only the occupation. There were two power peaks, one in the morning and one in the late afternoon, which likely corresponded to people arriving either to work in the morning or home in the afternoon. On weekends, these two peaks were not observed, since few people commute by car to work on weekends. Instead, a single peak can be seen on Saturday and Sunday, likely a result of people attending leisure activities.

The number of charging processes observed in early January was slightly below other periods. Unfortunately, it was not 100% clear whether this was a real effect or the result of data sampling. As a measure of caution, we would recommend that fellow researchers use the data from February as a representative shape instead. The scaling of the power demand was highly dependent on the amount of energy recharged per charge event. We advise researchers to compare these assumptions with their assumed vehicle energy demand.

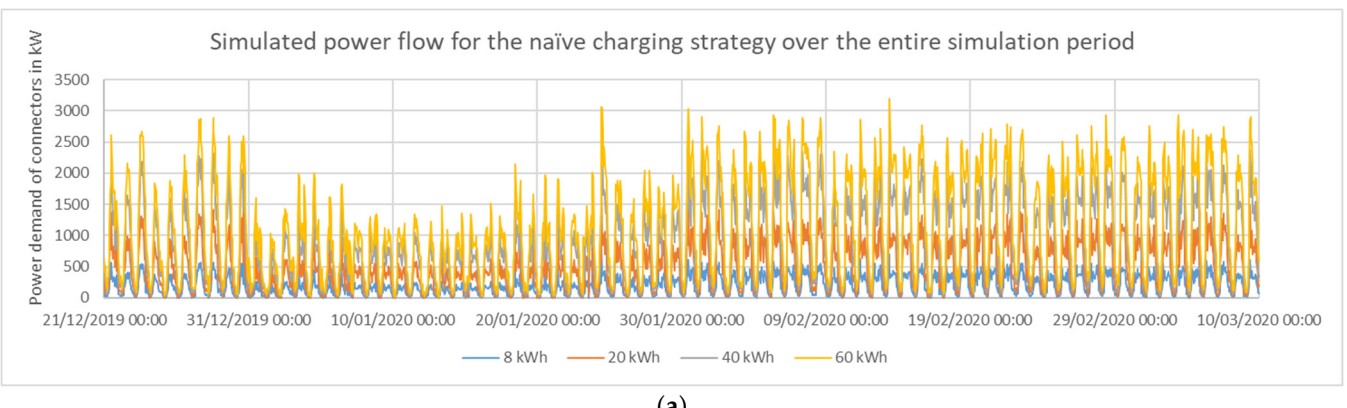

(**a**)

**Figure 9.** *Cont.*

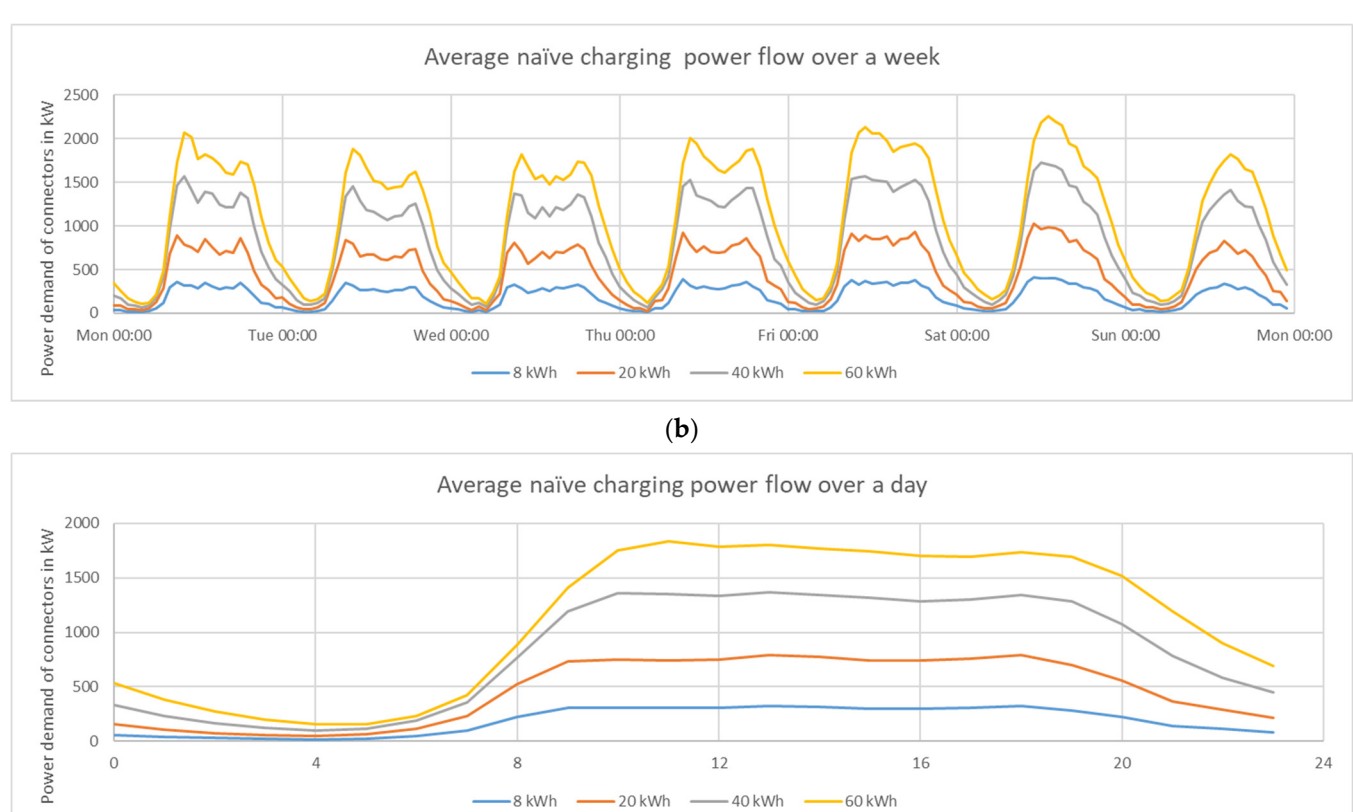

**Figure 9.** Power demand using the naive charging strategy for the 1562 connectors with a total power of ~34 MW observed in this study: (**a**) the power flow over the entire simulated timeframe; (**b**) the average of (**a**) over a week; (**c**) the average of (**a**) over a day. The plot is structurally very similar to Figure 5.

### 4.2. Peak Power Germany

An overview of the peak power demand of charging stations using the different strategies introduced earlier is shown in Figure 10 with details provided in Figures 11–14. Some of the key data points are highlighted below:

- The naive charging strategy led to only moderate peak power requirements of 2 MW for the observed 1562 connectors if 20 kWh were charged per charging process. If this number is extrapolated to the planned one million public connectors [37] planned for 2030, this would result in approximately 1.28 GW of additional power;

- Depending on the charged amount of energy, the peak power when using the unidirectional price-optimisation strategy or the $CO_2$ emission reduction strategy rises by a factor of 3 to 4 compared to the naive charging strategy. This was caused by the fact that all connectors where a vehicle is connected will charge at the cheapest/least emission intense timestamp, thereby causing a power peak. This effect was almost identically strong for both strategies;

- The peak power limitation strategy was able to flatten the power consumption during most observed hours. The peak power, however, did not change dramatically. Note that for the peak optimization algorithm, since the price-optimisation was run first, the peak power was above the peak power of the naive approach. This was the result of the optimization terminating as outlined in Section 3.1.4. The energy transferred during the peak hour, however, only represented <0.2% of the total energy transferred in all cases. It would consequently not cause great damage if a hard cap was introduced for that critical hour, since only few vehicles would be unserved. Additionally, it must

be noted here that the data collection process from public sources was not free of faults. It was not unlikely that the peak was a result of a data gap, where for some time new arriving vehicles were not detected and then were all counted as arriving at one instance in time. Correcting this would require better data quality, which unfortunately was not available for this study.

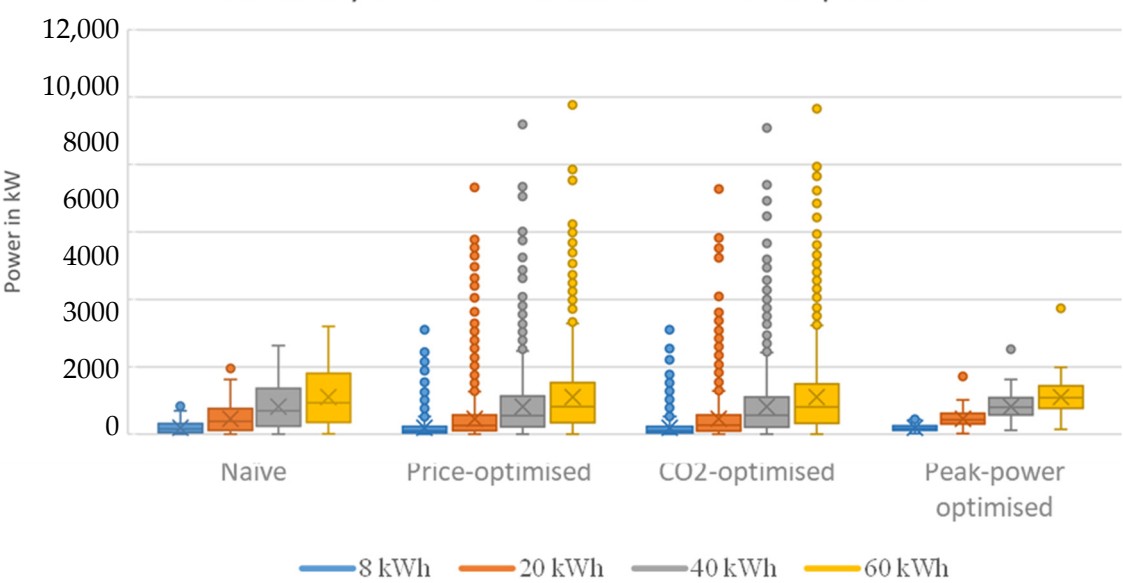

**Figure 10.** Summary box plot of the summed power values from Figures 11–14.

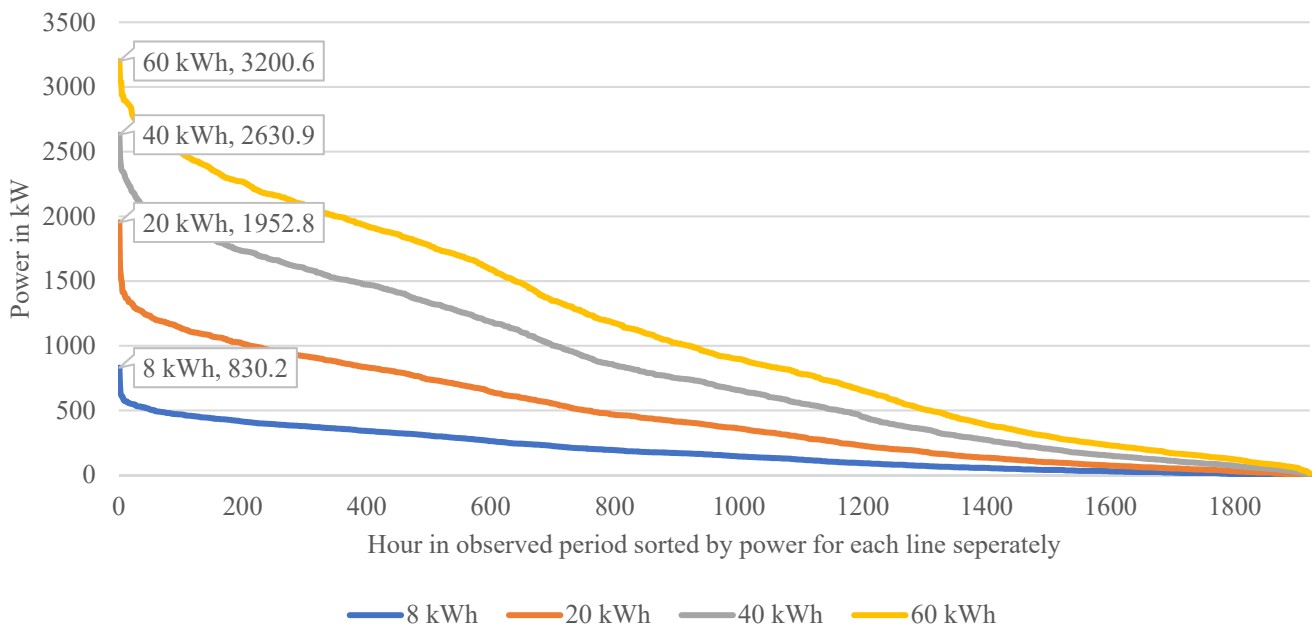

**Figure 11.** Sorted summed connector power when employing the naive strategy. For example, if during each charging process 40 kWh is to be charged, during ~800 h of the observed 1921 h, the chargers consumed 500 kW or less.

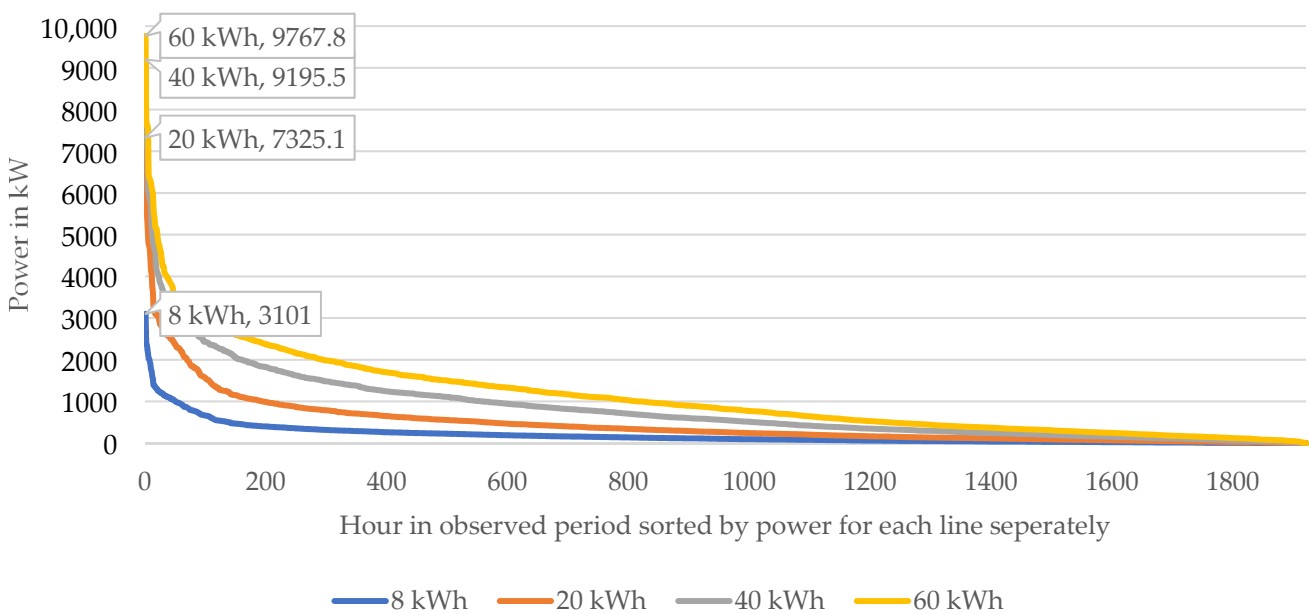

**Figure 12.** Sorted summed connector power when employing the price optimization strategy. For example, if during each charging process 40 kWh should be charged, during ~1400 h of the observed 1921 h, the chargers consumed 1000 kW or less.

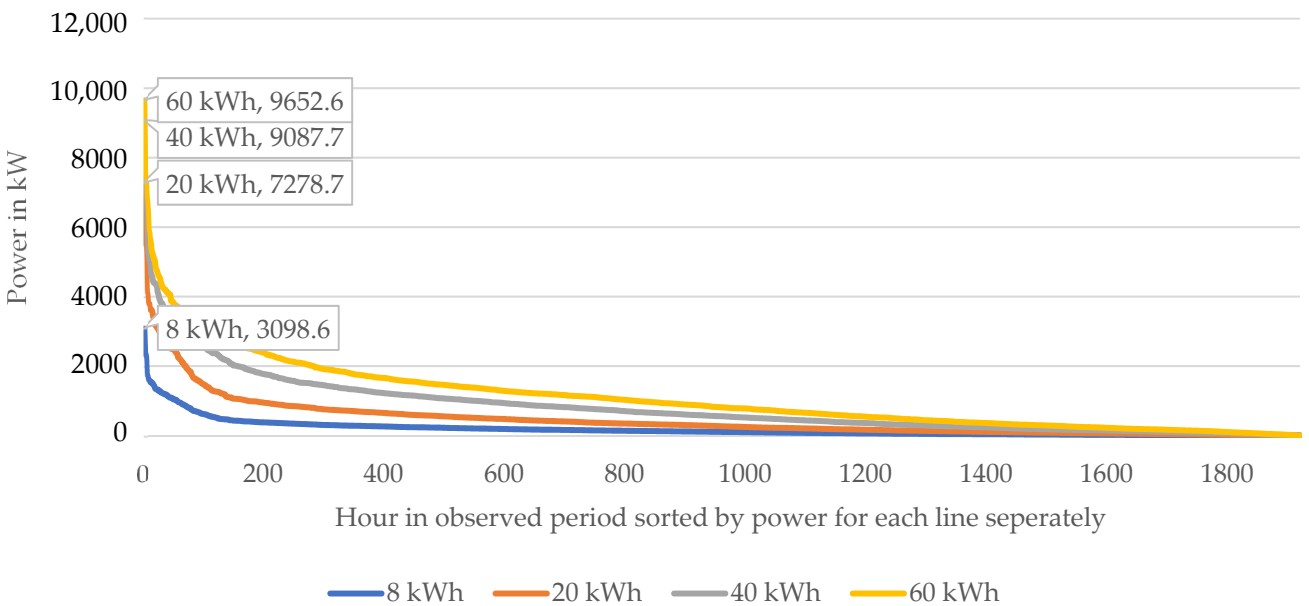

**Figure 13.** Sorted summed connector power when employing the $CO_2$ optimisation strategy. For example, if during each charging process 40 kWh should be charged, during ~200 h of the observed 1921 h, the chargers feed in power at a rating of 4000 kW or more.

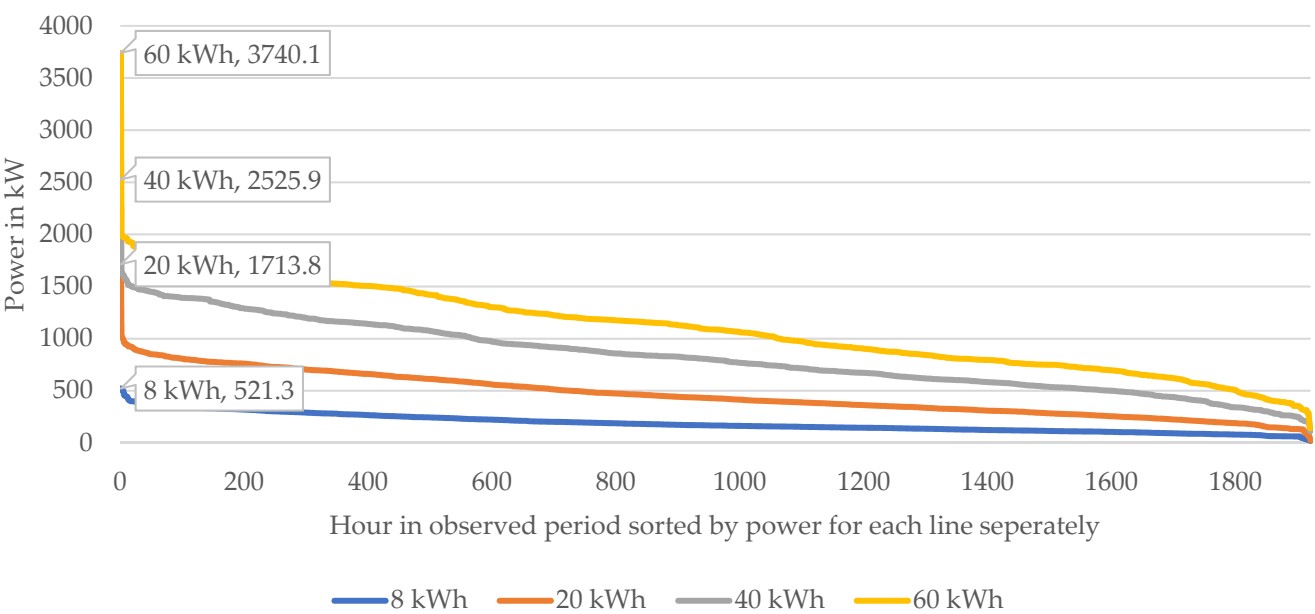

**Figure 14.** Sorted summed connector power when employing the peak minimization strategy. For example, if during each charging process 40 kWh should be charged, during ~400 h of the observed 1921 h, the chargers consumed 500 kW or less.

*4.3. Cost Comparison*

In the following, we analyse the costs incurred when using the different charging strategies on public charging infrastructure in Germany. The cost calculated as the multiplication of the energy consumption vector created in each optimization and the price vector divided by the total amount of energy charged.

$$\text{Cost} = \vec{E} * \vec{\lambda} * \frac{1}{\Sigma\left(\vec{E}\right)} \tag{6}$$

$$\vec{E} = \begin{pmatrix} E_1 \\ E_2 \\ \vdots \\ E_{1921} \end{pmatrix} ; \ \vec{\lambda} = \begin{pmatrix} \lambda_1 \\ \lambda_2 \\ \vdots \\ \lambda_{1921} \end{pmatrix} \tag{7}$$

where $E_x$ is the energy consumed by all C at hour, h; $\lambda_h$ is the wholesale power price taken for any given hour [32].

For the four main strategies, the costs are shown in Figure 15. Figure 16, in turn, shows how the average electricity costs changed if the various power limits defined in Section 3.2 were given to the peak power minimisation algorithm. The left end of the plot corresponds to the peak power minimisation strategy, since a goal peak power of 0 kW was given. Going along the *x*-axis to the right, the maximum allowed peak power steadily increased, which gave the optimiser more room to optimise for price instead of peak power. The price-optimised strategy corresponded to an allowed peak power of ∞. In practice, the right end in Figure 16 was almost identical to the price-optimised strategy already.

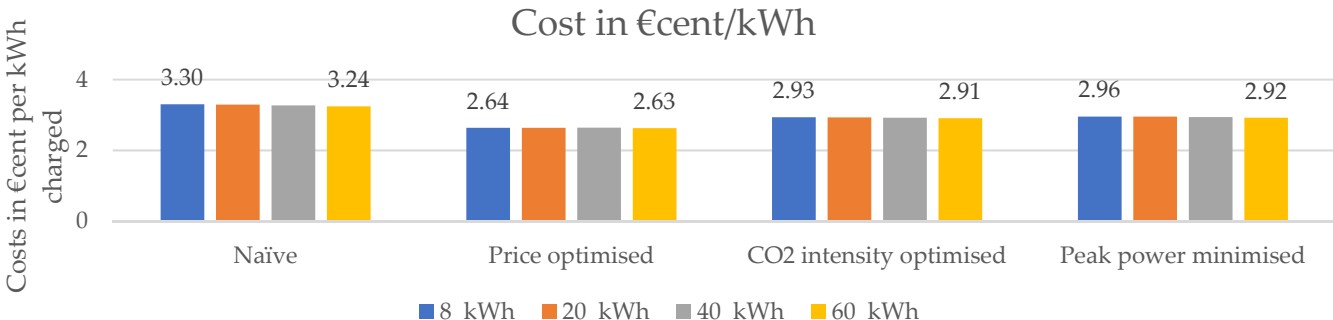

**Figure 15.** Costs in EURcent per kWh for the charged energy used assuming the energy recharged per charge event and charging strategies.

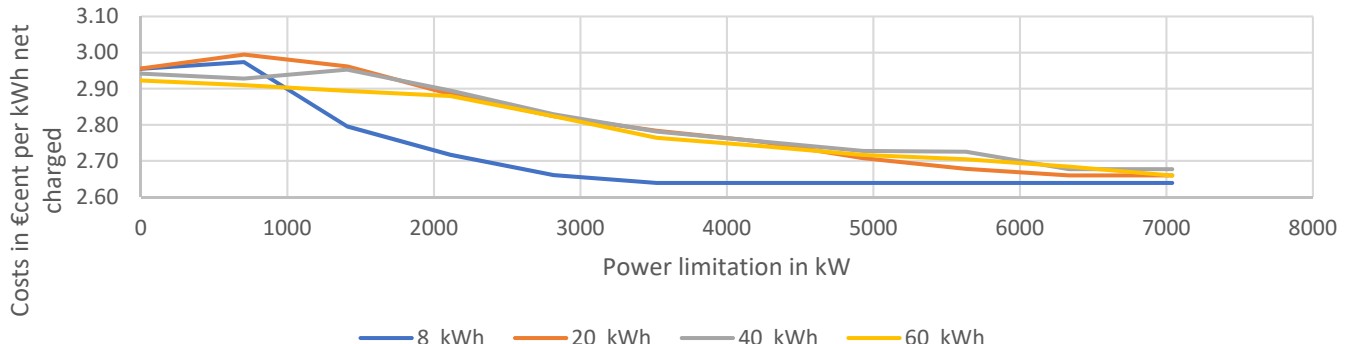

**Figure 16.** Costs in EURcent per kWh net recharged energy as a function of the global power limit for the power limits defined in Section 3.2. The left end is the same as "peak minimisation" and the right end is almost identical to the "price optimised" in Figure 15 as a little peak minimisation is required with a power limit of over 7040 kW.

Both figures show only the energy purchase cost. For the full customer price, factors such as grid usage fees, energy retail margin, taxes, levies, profit margins, etc., would have to be considered. The additional costs, however, are the same for each strategy, since the overall amount of purchased energy does not change.

Some key observations derived directly from the results shown in the figures are:

- The price differences between the naive strategy and the price-optimised strategy are smaller than 1 EURcent per kWh for the observed timeframe and all calculated amounts of energy per charge event;
- The electricity purchase cost per kWh is largely independent of the amount of energy charged per process. For all strategies, the price difference between the 8 kWh scenario and the 60 kWh scenario was less than 2%;
- Small price fluctuations exist for low power limits shown in Figure 16, as the optimization terminated due to the long runtime. Prices for a power limit of 0 kW were lower than for 704 kW, because each stricter power limit takes the results of the previous run as starting point. Since the optimization for 704 kW was eventually stopped, the new run had room left for improvement. The slight differences of less than 0.1 EURcent/kWh, however, did not change the overall picture;
- All in all, while the power increased by a factor of 3 between the naive and the price-optimizes strategy, the considered costs only decreased by 20%.

### 4.4. $CO_2$ Intensity

The $CO_2$ intensity was calculated in a similar fashion compared as the electricity costs by replacing hourly electricity costs with grid $CO_2$ intensity taken from [33]. Figure 17 shows how the different strategies perform in terms of $CO_2$ intensity. We do not include local renewable generation in our calculation which is nowadays already frequently com-

bined with charging infrastructure [38]. This is due to the fact that the generation would otherwise be fed into the grid and replace flexible generators such as gas or coal fired power plants. Some key results that can be concluded from Figure 17 are:

- Price-optimisation already reduced $CO_2$ emissions compared to the naive base case. This is likely a result of the correlation already shown in Figure 6c. Renewable generators reduce the wholesale electricity price since their variable costs are often 0 €. Optimising for price consequently means optimising for renewable, $CO_2$-free electricity in many cases;
- The $CO_2$ intensity was overall much lower than the time-weighted average in Germany of 324.20 g $CO_2$/kWh and the consumption-weighted average of 326.37 g $CO_2$/kWh;
- The $CO_2$ intensity optimisation was able to reduce the $CO_2$ intensity of charge electricity by ~6.6%.

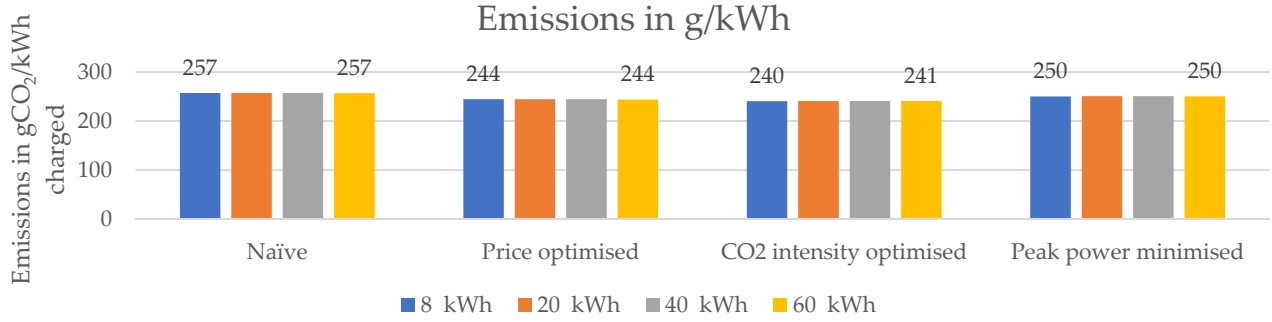

**Figure 17.** $CO_2$ intensity of the electricity charged from the grid using assumed energy recharged per charge event and charging strategies.

### 4.5. Simultaneity Factor

The SF was calculated for sets of connectors with the numbers of connectors in the below set. Since our focus was on connectors, the definition of the SF was the same as in Figure 3.

$$S = \left\{ x : x = a*10^b \wedge a \in [1, 2, \ldots 10] \wedge b \in [0, 1, 2] \right\} \tag{8}$$

For each element n in S, a set $S_n$ with 1000 elements was created by randomly selecting n connectors each and ensuring that each set is unique.

$$S_n = \left\{ x : x_i = F \wedge x_i \notin [x_1, x_2, \ldots x_{i-1}] \wedge i \in [1, 2, \ldots 1000] \right\} \tag{9}$$

$$F = \left\{ y : y_j \xleftarrow{R} C \wedge y_j \notin \left[ y_1, y_2, \ldots y_{j-1} \right] \wedge j \in [1, 2, \ldots n] \right\} \tag{10}$$

The result for the SFs given the different strategies (naive, price-optimised, bidirectional price-optimised, peak power limitation) and energy demands per CE (8 kWh, 20 kWh, 40 kWh and 60 kWh) are shown in Figures A3–A6 with a summary given in Figure 18. The key results of these figures are given in the following:

- Already for groups of 10 connectors, the SFs reduce to about 50% compared to the value for a single connector;
- For individual connectors, the SF is 1 in virtually all cases. All connectors were consequently used at least once in the observed period;
- The amount of energy required has a large impact on the SF of larger sets of connectors, particularly for the naive and the peak-power strategy. In these two, larger amounts of energy required lead to overall longer charging times which in turn increases the likelihood of overlaps between charging processes;
- In terms of SF, there is little difference between price- and $CO_2$-optimised strategy. This was to be expected since both optimisation function in a very similar fashion;
- For the price- and $CO_2$-optimised strategy, the SF does not change as strongly between the different amounts of energy charged. This is because the peak power occurs at the

time step with the lowest price or lowest $CO_2$-intensity. The optimiser will move as much charging as possible to this critical hour. This process therefore unaffected by the length of the individual charging processes;

- Even in the extreme cases of vehicles recharging much energy or the simplified bidirectional charging, large sets of connectors seldom exceed an SF of 40%;
- The difference between the price-optimised and non-price-optimised strategies is particularly prominent for 8 kWh of recharged energy and for smaller sets of less than 100 connectors;
- For the largest number of connectors considered, the SF varies around 20% for the naïve strategy. This shows, that the EVs are not going to cause dramatic grid overloads by simultaneity.

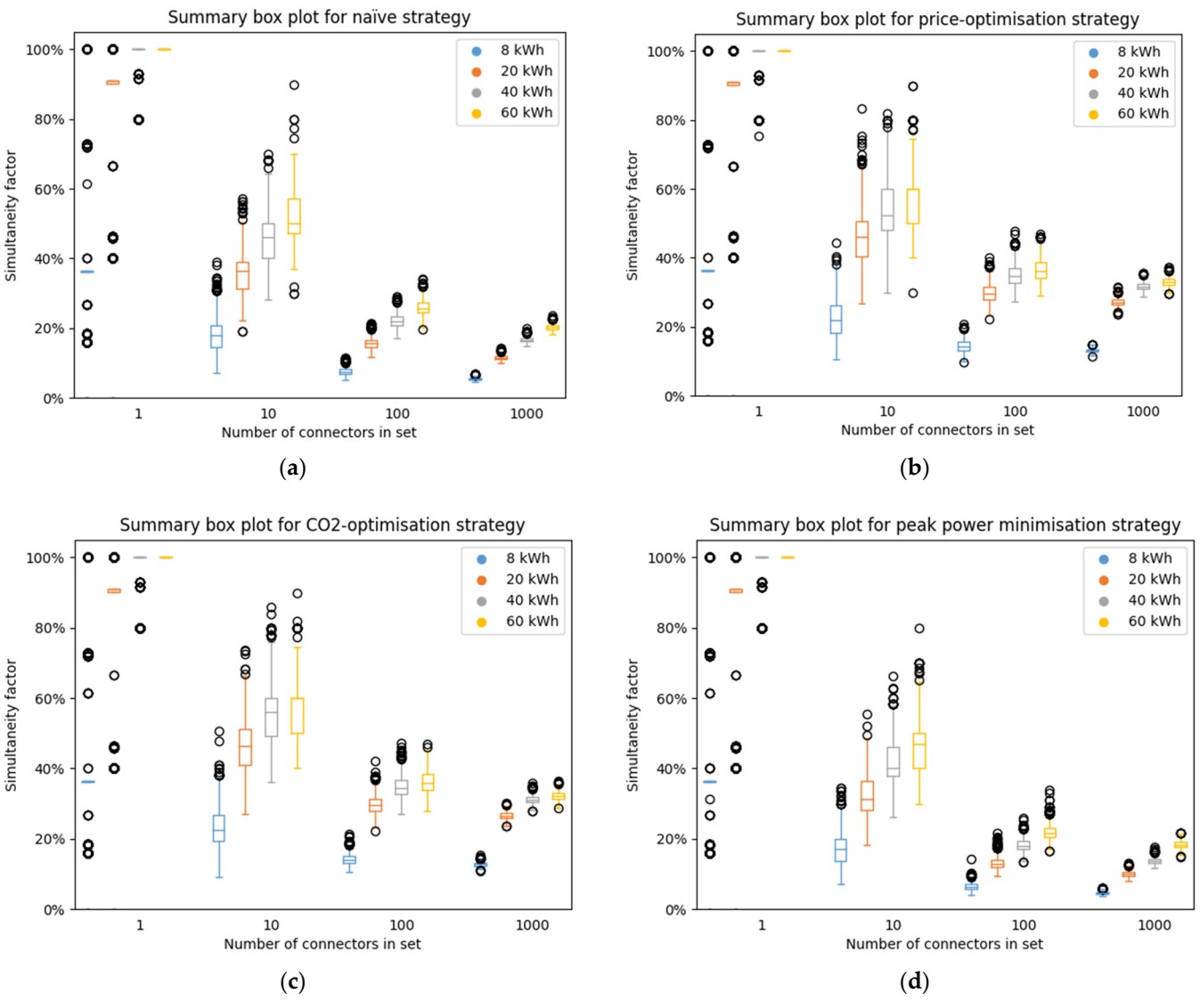

**Figure 18.** Summary of the simultaneity factors assuming for (**a**) naive charging; (**b**) price-optimised charging; (**c**) $CO_2$ intensity optimised charging; (**d**) peak power optimised charging. The plots summarise the information given in Figures A3–A6. For each combination of number of connectors per set and energy per charge event, 1000 sets were randomly created, and the results displayed in the box and whiskers plots in this figure. For example, when applying the $CO_2$-optimisation strategy, a random selection of 100 connectors lead to a median simultaneity factor of 30% if a 20 kWh per charge event were assumed with outliers ranging from 20% to 42%.

## 5. Discussion

As outlined in the introduction, this paper aims to support policy makers, grid companies and utilities in designing their policies and regulations regarding recharging strategies when using (public) charging infrastructure. The results show that different recharging strategies have a strong influence on peak grid usage and thereby confirm theoretical findings from literature. By using real-world data, this paper strengthens the validity of the previously mostly theoretic discussions. The found results are also in line with field tests using smaller vehicle fleets [21,39]. Netze BW, for instance, reported that for a fleet of 50 vehicles and 34 connectors, the maximum simultaneity factor was 25% [39], which is in line with our 20 kWh naive charging strategy scenario. Similarly EnBW reported 4 out of 10 vehicles charging in parallel as maximum [21] which again fits to the 20 kWh naive charging strategy scenario.

What has to be kept in mind when working with the results is that we excluded connectors that were never used. Especially in certain rural areas, there might be a significant share of unused assets. We however do not believe that unused connectors should be completely ignored in planning grid capacities. These unused connectors are consequently wasting money both by never being used, but also by occupying grid capacity that is used only seldom.

### Limitations

To handle the large amounts of data, the methodology used in this paper made some simplifications that the reader should be aware of when working with our results. The main limitations are given below:

- Hourly resolution

  All calculations in this paper were performed with an hourly resolution and only for the average power consumption observed during each hour. Most charging processes took significantly longer than 1 h [2], and it is therefore acceptable to use this resolution. For very short processes, an inaccuracy was introduced. This might slightly lower peak power demand as reported in Section 4.1 and SF for larger sets as reported in Section 4.5. If two short processes occur in the same hour but do not overlap, they would appear as overlapping in this study.

- Simplified vehicle charging model

  Since no information about the arriving vehicles was present in this study, the assumption was made that the vehicle was able to charge at the power level dictated by the charging station. Many car models today are not yet able to charge at 22 kW.

## 6. Conclusions

EVs can be a game-changer in the shift to a renewable energy system, since they represent highly controllable loads. Grid operators are currently challenged with the question of how to plan electric grids given that the growth rates and electricity demands of EVs are hard to predict. This paper supports this discussion by providing insights into how charging strategies affect demand. Several conclusions from the analysis are given below.

### 6.1. SFs for Public Charging Infrastructure Were Found

In this paper, SFs for differently sized sets of charging station connectors were calculated. Given the found result that ten connectors had half the mean SF compared to a single connector, grid operators have a basis for planning grid capacities. The standard deviations in the calculated cases are also reasonably narrow. This indicates that even if grid capacities are insufficient in rare cases and the power flow would have to be curtailed, the loss of comfort for users is likely minimal and such events would happen only rarely. Our results are slightly below current recommendations for grid planning in Austria [16] that were shown in Figure 3. The Austrian recommendations also include home-charging where a higher simultaneity factor could be expected as in the case of public charging;

### 6.2. Naive Charging Is Challenging Only on the Distribution Grid Level

On the aggregated German level, naive charging in 2030 would create around 1.28 GW of additional peak load. Compared to currently ~80 GW peak load, the impact of EVs on the transmission system is manageable. On the distribution grid level, smaller hotspots might be created since SFs of 30% to 40% are possible for groups of 100 connectors.

### 6.3. Price-Optimisation Can Cause Local Hotspots

Price-driven recharging schemes need to be coordinated well with local grid operators since SFs increase significantly for small groups of connectors that all charge according to national prices. Possible solutions could be to include a dynamic grid usage fee that balances the benefits from cheaper electricity purchase costs with requirements of the local grid. Other options are to pair local generators such as photovoltaics with charging infrastructure. This would align incentives of grid operator and station operator since using locally generated energy is most cost-efficient for the operator and simultaneously reduces grid stress.

### 6.4. Wholesale Electricity Price Optimisation Did Not Reduce Electricity Purchase Costs Significantly

The difference between energy costs using the price-optimised strategy and the peak power reduction strategy was on the order of 0.3 EURcent/kWh and, consequently, insignificant compared to other cost factors such as grid fees, taxes, or levies. This could be different, if for instance electricity from onsite PV-generators could be used. Since public charging stations typically do not have space available to do so, this option is more viable for private charging infrastructure.

### 6.5. Local Storage Units Need to Be Used Intelligently

Particularly fast-chargers are often coupled with buffer battery storages that can provide power to vehicles even if the local grid is overloaded. If these battery storages are used for arbitrage trading during off-peak hours, care should be taken to not augment hotspots. If storages and vehicles would both charge based on grid price or $CO_2$ intensity, their combined effect could be detrimental for grid stability.

### 6.6. The Introduced Algorithms Are Fast and Efficient

All algorithms are efficient and achieve their goal while keeping the number of lines of code low and allowing for vectorised operation. A key challenge is the balance between price optimisation and peak power reduction. Price optimisation creates peaks that then need to be reduced. This might cause peak powers for the peak power optimisation to be higher than they would have been if another starting configuration such as the naive charging algorithm were chosen.

**Author Contributions:** C.H.: conceptualization, methodology, software, validation, formal analysis, investigation, resources, data curation, writing—original draft, writing—review and editing, and visualization; J.F.: writing—review and editing, supervision, and project administration; D.U.S.: supervision, project administration, and funding acquisition. All authors have read and agreed to the published version of the manuscript.

**Funding:** The work presented in this paper was created in the context of and supported by the project "ALigN—Ausbau von Ladeinfrastruktur durch gezielte Netzunterstützung" funded by the Federal Ministry for Economic Affairs and Climate Action of Germany according to a decision of the German Federal Parliament (01MW18006G).

**Institutional Review Board Statement:** Not applicable.

**Informed Consent Statement:** Not applicable.

**Data Availability Statement:** The summarized result datasets shown in Figures 9–18 are available from the corresponding author upon reasonable request. The input data containing the occupation status of all charging points also used in [2] cannot be shared, as the contained information is confidential. As a consequence, the detailed results of when which charging point was active also cannot be shared.

**Acknowledgments:** We would like to thank our colleagues who have given valuable feedback in the creation of this paper.

**Conflicts of Interest:** The authors declare no conflict of interest.

**Nomenclature**

| | |
|---|---|
| Connector, C | A socket or cable at a charging station to which an electric vehicle can be connected |
| Charge event, CE | A charge event starts when an electric vehicle is connected to a connector and ends once it is disconnected again |
| Power limit | The power limit is the power available globally for all connectors |
| Power capacity | The rated power of a connector |
| EV | Electric vehicle |
| SF | Simultaneity factor |

**Appendix A. Pseudo Code**

The following two figures show the pseudo-code of the two algorithms used in this paper. The syntax is similar to Python.

```
1 demand = [energy required per charge event]
2 power = [power capacity per connector]
3 sorted_index = timesteps.sortby(price,asc)
4 for ts in sorted_index:
5   events = all charging events active at ts
6   chrg = [energy already charged per event]
7   remain = demand[events] – chrg
8   for each event:
9       if remain[event] > power_limit_sttn:
10        Charge at full power
11      else:
12        Charge remain[event]
```

**Figure A1.** Price-optimisation pseudo code.

```
1 no_improv = []
2 while len(no_improv) < len(sorted_index):
3   crit_ts = maxidx(power) not in no_impro
4   if power[crit_dt] <= power_limit:
5       break
6   distribute power away from crit_ts to cheapest alternative
7   if no option to distribute away:
8       no_improv.append(critical hour)
```

**Figure A2.** Peak power reduction pseudo code.

## Appendix B. Simultaneity Factor for the Different Scenarios and Strategies

The following figures show the simultaneity factor for the different strategies and scenarios as discussed for instance in the results Section 4.5.

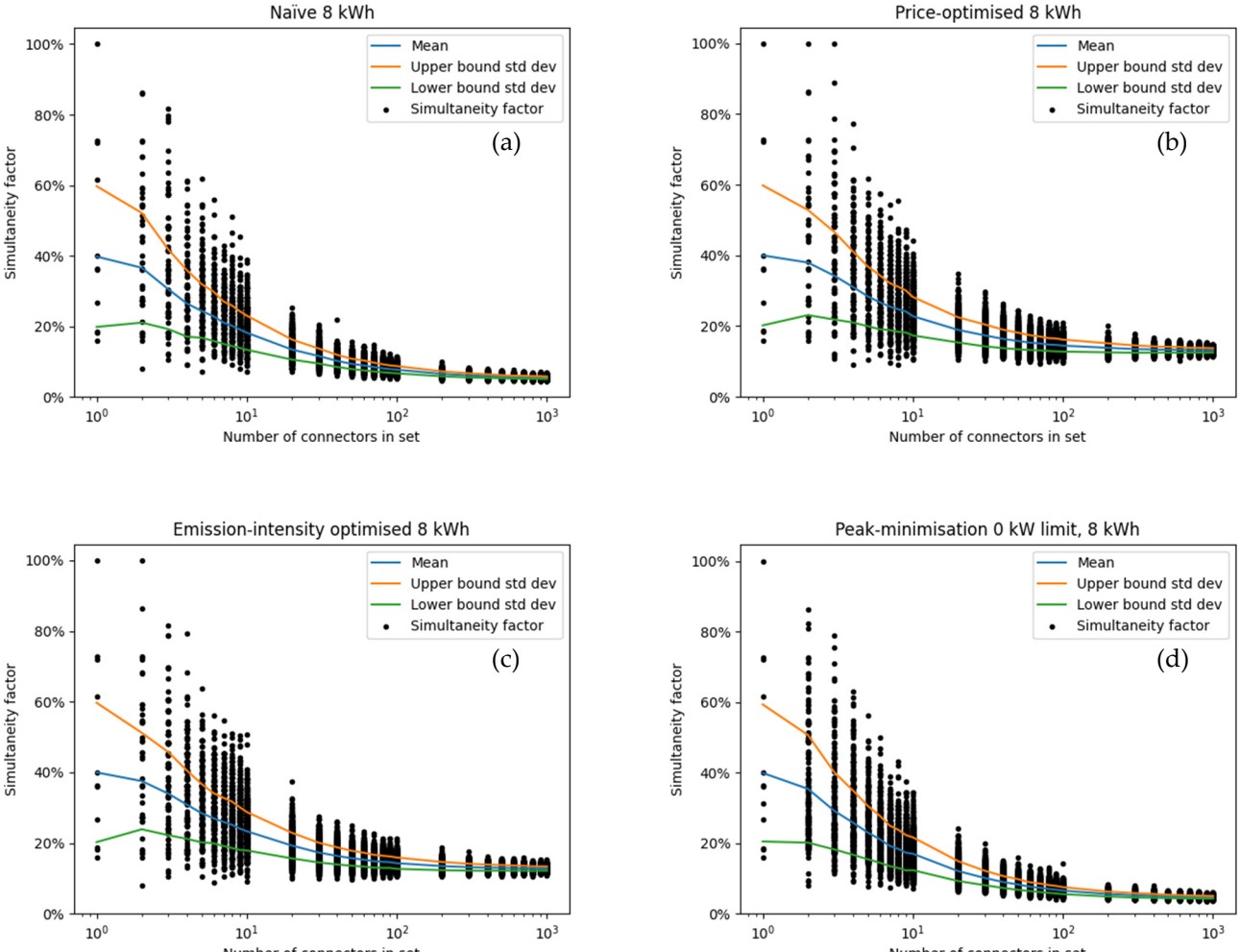

**Figure A3.** Coincidence factor assuming 8 kWh per CE for (**a**) naive charging; (**b**) price-optimised charging; (**c**) $CO_2$ intensity optimised charging; (**d**) peak power optimised charging. Note that peak power was optimised for all C simultaneously. Optimizing for each set in (**d**) for a low simultaneity factor likely results in significantly lower values. For the larger sets, the difference is minor. The blue line shows the mean across the randomly selected sets, and the orange and green lines the upper and lower boundaries of the standard deviation. Each randomly selected set was of the size shown on the *x*-axis.

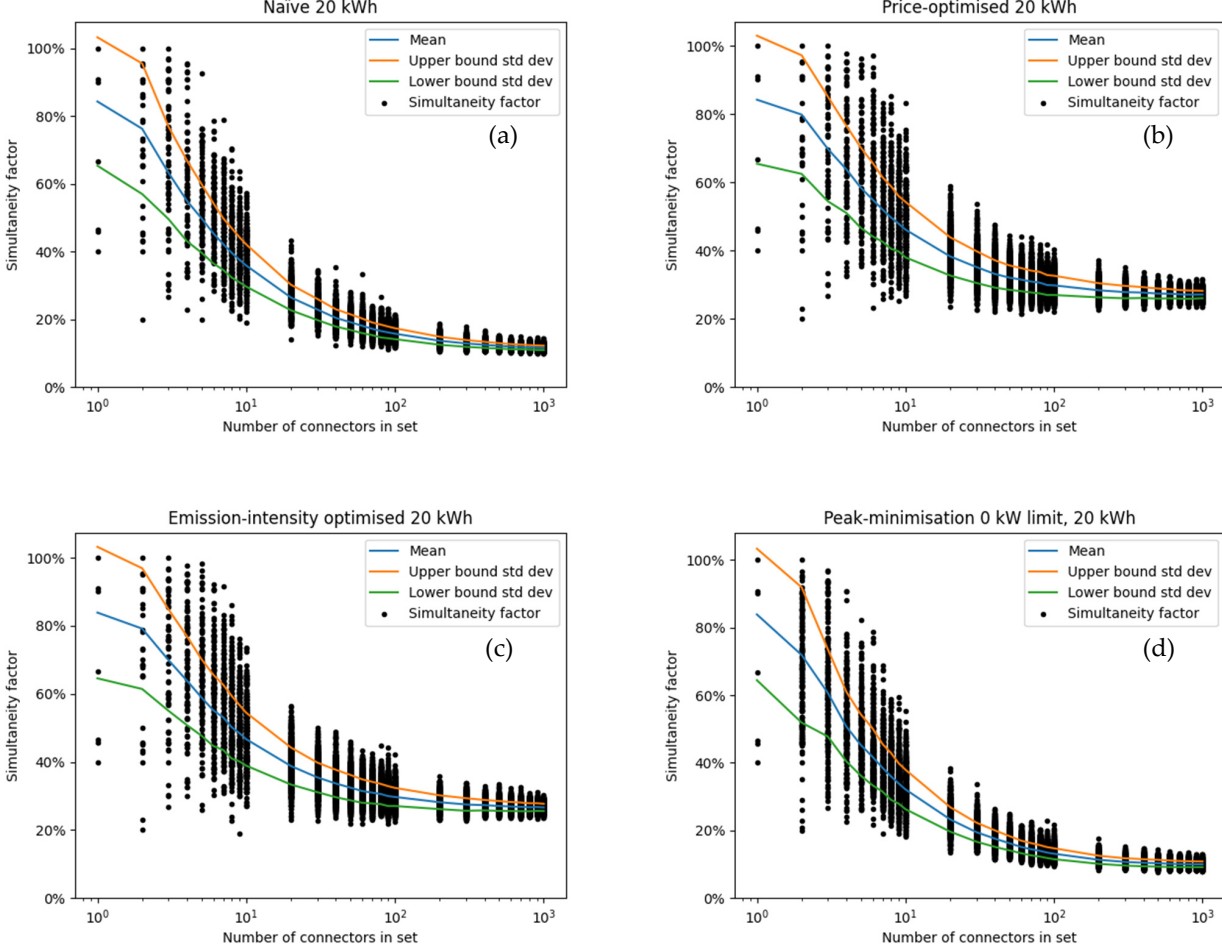

**Figure A4.** Coincidence factor assuming 20 kWh per CE for (**a**) naive charging; (**b**) price-optimised charging; (**c**) $CO_2$ intensity optimised charging; (**d**) peak power optimised charging. Note that peak power was optimised for all C simultaneously. Optimizing for each set in (**d**) for a low simultaneity factor likely results in significantly lower values. For the larger sets, the difference is minor. The blue line shows the mean across the randomly selected sets, and the orange and green lines the upper and lower boundaries of the standard deviation. Each randomly selected set was of the size shown on the *x*-axis.

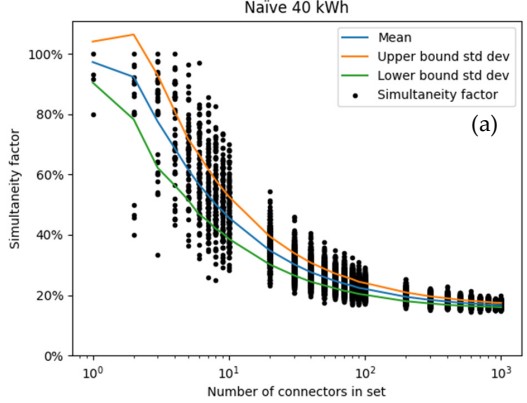

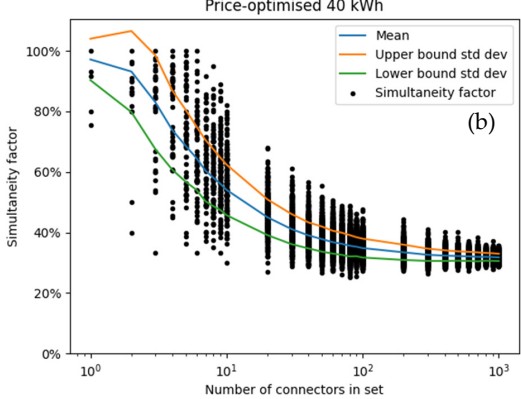

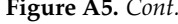

**Figure A5.** *Cont.*

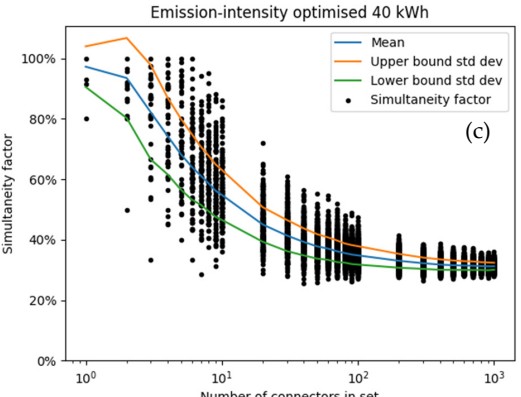
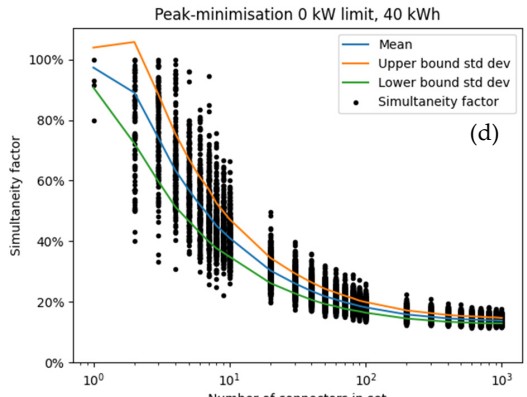

**Figure A5.** Coincidence factor assuming 40 kWh per CE for (**a**) naive charging; (**b**) price-optimised charging; (**c**) $CO_2$ intensity optimised charging; (**d**) peak power optimised charging. Note that peak power was optimised for all C simultaneously. Optimizing for each set in (**d**) for a low simultaneity factor likely results in significantly lower values. For the larger sets, the difference is minor. The blue line shows the mean across the randomly selected sets, and the orange and green lines the upper and lower boundaries of the standard deviation. Each randomly selected set was of the size shown on the *x*-axis.

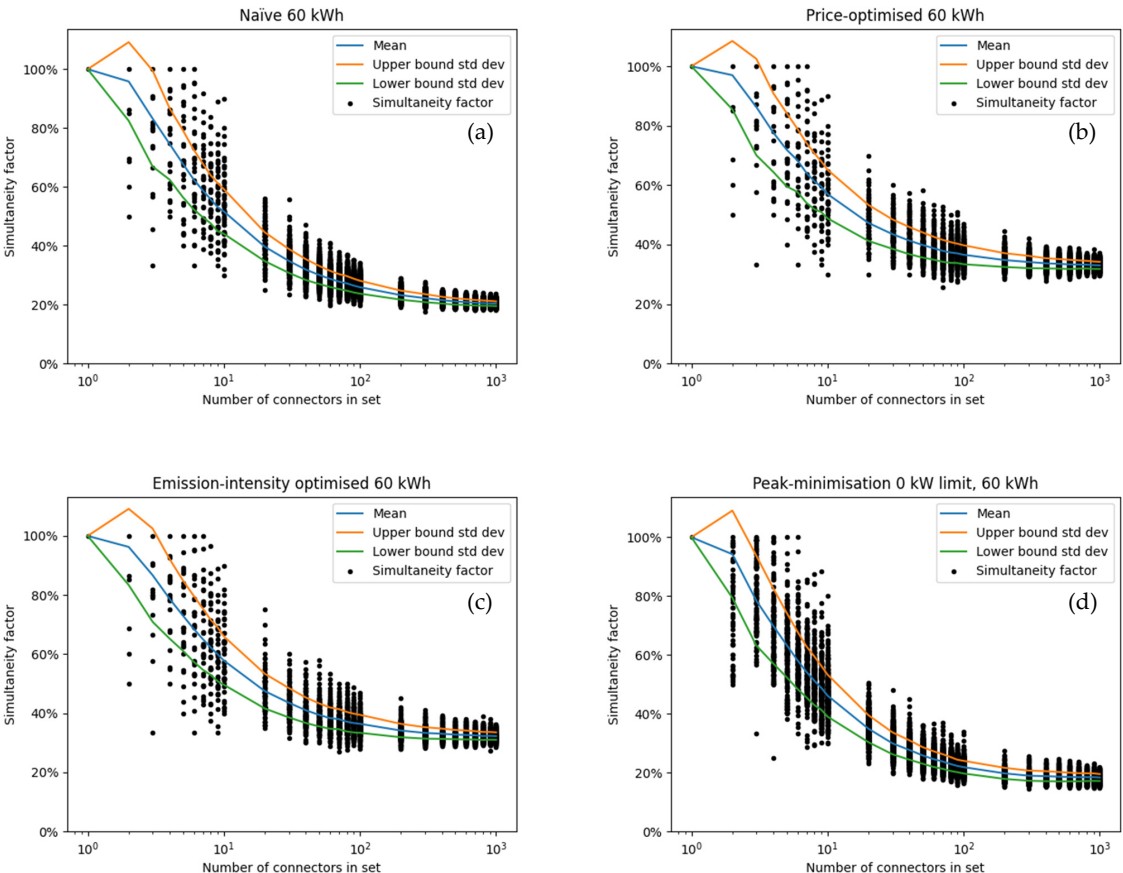

**Figure A6.** Coincidence factor assuming 60 kWh per CE for (**a**) naive charging; (**b**) price-optimised charging; (**c**) $CO_2$ intensity optimised charging; (**d**) peak power optimised charging. Note that peak power was optimised for all C simultaneously. Optimizing for each set in (**d**) for a low simultaneity factor likely results in significantly lower values. For the larger sets, the difference is minor. The blue line shows the mean across the randomly selected sets, and the orange and green lines the upper and lower boundaries of the standard deviation. Each randomly selected set was of the size shown on the *x*-axis.

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
