# Peer review of "Simultaneity Factors of Public Electric Vehicle Charging Stations Based on Real-World Occupation Data"

_wevj, doi:10.3390/wevj13070129_

Round 1

Reviewer 1 Report

An article Simultaneity factors of public electric vehicle charging stations 2 based on real-world occupation data. It is presented and finally it is well validated.  I think the article needs minor changes because:   1. In the abstract, the purpose and main contribution of the article are not clearly stated, and there is no main need for the purpose of the research.   2. The following sources should also be used in the introduction. Article innovation and article structure It is very appropriate to read the following articles. 1-DOI: 10.1049/iet-rpg.2020.0399 2-DOI: 10.1109/ICEE50131.2020.9260941.   10.1109/ICEE50131.2020.9260941. https://doi.org/10.3390/wevj12040190.    3. I think the presented method needs more preliminary explanations for the reader's understanding.   4. The conclusion needs to be rewritten.   And resources should be updated. And the shapes should be shown bigger if possible and the interpretation about the lines should be provided

Author Response

Thank you very much for your kind review and constructive comments. Please find our responses to your statements below:

1: The abstract was adjusted
2: Source 1 and 3 were added in an appropriate location. Souce 2 was not added since it refers to the vehicle's internal energy management system which is not part of this analysis
3: A paragraph with preliminary explanations was added at the beginning of the methods section
4: The conclusion was revisited and adapted in a few locations. Unfortunately, it is not clear from the comment alone what the exact wishes of the reviewer are.
5: Images are as large as the layout permits and it will be possible to enlarge them in the web version. We therefore hope that the readibility is ensured. Where applicable we also added explanations to the figure legend to aid in the interpretation of the lines.

Reviewer 2 Report

Simultaneity factors of public electric vehicle charging stations based on real-world occupation data

The article is very well written. It explains the impact of EV charging strategy on grid and evaluates five different parameters which justifies the proposed methodology. Literature is well explored, and contribution is very clear. This paper should be accepted for publication after some minor revisions if possible (not compulsory).

Below are some recommendations.

1.     Impact on electricity cost has been taken from [3] as justification of research gap. It would be better if the said research is analyzed for three different EV charging scenarios i.e., high density charging when most of connectors are occupied, medium density and low density.

2.     Flow chart or more detail of optimization algorithms, especially price optimization algorithm if added, will be better.

3.     In figure 7, price/charging event is low in some cases. What factors are involved in that specific cases?

4.     Detail of CO2 emission optimization is required with brief analysis.

5.     During high power demand hours, which randomly changes hourly, what is the strategy of utilities to meet power demands? Explain this factor in terms of starting extra generators and also analyze economic impacts of peak demand hours.

6.      Figure 18 requires explanation.

7.     In section 6.1 limitations are irrelevant. EVs almost takes 2-4 hours for CE. Hourly comparison should be replaced with average power consumption per hour.

8.     In sec 7.1 recommendation cited in [16], if descriptively explained, will be better.

9.     “Price optimization can cause local hotspot”. What is the proposed recommendation for utility company to overcome the said conclusion?

10.  Flow chart of the proposed methodology of the whole article if added to article will make it understandable quickly. The research conducted is very well explained but it need pictorial representation.

Author Response

Thank you very much for the kind review. A few notes on the optional content that you suggested:

  1. We intentionally decided against doing so since we did not want to artificially modify the data. You can however derive these scenarios by picking different combinations in Figure 7.

  2. More details were added.

  3. The numbers are illustrative only and no specific reason is given. The caption was extended to make this more clear.

  4. The explanation was slightly extended. Given that it is essentially identical to the price optimisation however, we do not want to add a very extensive explanation for the sake of conciseness.

  5. A new section 7.5 was introduced to discuss this.

  6. An example how to read the figure was included in the caption.

  7. We kept the title of the limitations section, but emphasized that this leads to the average power consumption.

  8. A link to the figure showing the recommendations was included in section 7.1

  9. The paragraph following this statement contains suggested solutions.

  10. We created a graphical abstract that should fulfil the function that you describe.

Reviewer 3 Report

The authors have presented a very good analysis on the impact of simultaneity on the EV charging stations performance. The authors can incorporate the following suggestions for improving the readability of the paper .

1. The authors need to give the citations for the optimization algorithms they have used in Section-4.

2.  The authors have to provide the mathematical expressions of the objective functions used in Section-4.

3. The authors have to compare efficiency of the four optimization techniques  to handle simultaneity.

Author Response

Thank you very much for the kind review.

Following your suggestions, the optimization goal was formulated as a mathematical equation for sections 4.1.2 and 4.1.4. Since 4.1.1 is not optimized, no such equation is provided and 4.1.3 is almost identical to 4.1.2 and the difference was therefore described in words.

The algorithms are of our own making and no source is therefore necessary.

The efficiency is compared in the results section, but we have to emphasize here that only the last algorithm actually has the goal of reducing simultaneity while all other ones pursue other goals which actually lead to higher simultaneities.